# Unlocking the Secret to Higher Crop Yield: The Potential for Histone Modifications

**DOI:** 10.3390/plants12081712

**Published:** 2023-04-20

**Authors:** Weiwei Fang, Carlo Fasano, Giorgio Perrella

**Affiliations:** 1Department of Biosciences, University of Milan, Via Giovanni Celoria 26, 20133 Milan, MI, Italy; weiwei.fang@unimi.it; 2Trisaia Research Center, Italian National Agency for New Technologies Energy and Sustainable Economic Develoment, (ENEA), 75026 Rotondella, MT, Italy; carlo.fasano@enea.it

**Keywords:** histone modifications, rice, tomato, stress resistance, crop yield improvement

## Abstract

Histone modifications are epigenetic mechanisms, termed relative to genetics, and they refer to the induction of heritable changes without altering the DNA sequence. It is widely known that DNA sequences precisely modulate plant phenotypes to adapt them to the changing environment; however, epigenetic mechanisms also greatly contribute to plant growth and development by altering chromatin status. An increasing number of recent studies have elucidated epigenetic regulations on improving plant growth and adaptation, thus making contributions to the final yield. In this review, we summarize the recent advances of epigenetic regulatory mechanisms underlying crop flowering efficiency, fruit quality, and adaptation to environmental stimuli, especially to abiotic stress, to ensure crop improvement. In particular, we highlight the major discoveries in rice and tomato, which are two of the most globally consumed crops. We also describe and discuss the applications of epigenetic approaches in crop breeding programs.

## 1. Introduction

In the natural environment, plants are exposed to constantly changing stimuli; some of these changes follow regular natural cycles, such as seasonal change and the circadian clock, while others are severe and unfavorable. As sessile organisms, plants have evolved sophisticated mechanisms to increase their adaptation to rapidly changing conditions. Together with the genetic signaling network, epigenetic mechanisms are also considered critical in enhancing plants’ growth, development, and adaptation to adverse environmental stimuli [1,2,3,4]. Epigenetic modifications lead to heritable altered phenotypes by modulating gene expression through changes in chromatin accessibility, without affecting DNA sequences [5,6]. Epigenetic markers, including DNA methylation, non-coding RNAs, and histone modifications, all function to regulate chromatin structure and the subsequent gene expression [7]. DNA methylation primarily occurs in three sequence contexts—CG, CHG, and CHH—and plays a role in repressing gene expression [8]. Non-coding RNAs mediate RNA silencing that affects protein structure [9]. Both of these two markers are responsible for regulating gene transcription and are significantly involved in various developmental transitions. A more detailed introduction to DNA methylation and non-coding RNA mechanisms can be referenced in [4,10]. Histone modifications specifically refer to the post-translational modifications on N-terminal tails, including but not limited to methylation, acetylation, phosphorylation, ubiquitination, glycosylation, and sumoylation [11], that ultimately may determine chromatin configuration (open or closed) and regulate gene transcription [6]. Previous studies demonstrated that histone modifications, especially methylation and acetylation, are critical in mediating crop development. For instance, in rice, flowering efficiency and grain size are two main factors that determine yield [12], and they are under the regulation of histone modifications. Likewise, recent epigenetic studies in tomato are mainly focused on fruit ripening and flavor [13]. In addition, abiotic stresses, including drought, extreme temperature (heat and freezing), heavy metal, and excessive soil salinity constrain plant growth and crop yield [2]. Thus, it is a primary aim to improve crop resistance to various adverse conditions. Histone modifications were found to participate in the regulation of plant reproduction, fruit quality, and stress tolerance. This review focuses on recent progress in understanding the role of histone modifications with regard to rice and tomato yield and development. Indeed, the epigenetic regulatory mechanisms underlying enhanced growing quality provide additional approaches for crop improvement to cope with the increasing challenges for plants growth, as well as to meet the growing demand for food supply.

## 2. Histone Modifications

In eukaryotic cells, the basic unit of chromatin is the nucleosome, formed by two copies of histones (H3, H4, H2A, and H2B) and wrapped by 145–147 bp of DNA [14]. The chromatin’s conformation undergoes reversible modification, which corresponds to the transcriptionally inactive and active stage, respectively [15]. Epigenetic markers modulate gene expression by controlling chromatin conformation reversion and accessibility.

Histone modifications are reversible processes under the regulation of enzymes that either catalyze or remove specific marks termed as writers and erasers, respectively. Additionally, these marks are annotated by reader proteins, which give rise to subsequently altered phenotypes [16]. Both the specificity and the position along the histone tails determine the positive or negative regulation on gene expression [2].

The most used methodology for determining and quantifying the enrichment of specific histone markers upon DNA regions is chromatin immunoprecipitation (ChIP). ChIP assays are based on the use of antibodies that recognize particular histone modifications along DNA fragments [17,18]. However, over the last decade, plant scientists have developed technologies that allow for the capture of chromatin changes and genome architecture at different and multiples scales [19,20]. This includes high throughput chromosome conformation capture (Hi-C), also combined with ChIP (HiChIP), chromosome conformation capture (3C) which, combined with imaging methods, allows for the elucidating of chromosome territories (CTs), A/B compartments, topologically associated domains (TADs), and chromatin loops [21,22,23,24,25,26].

Similarly, the assay for transposase-accessible chromatin combined with sequencing (ATAC-seq) is a novel technology that can identify cis-regulatory elements and enhancer regions in plants [27,28].

### 2.1. Histone Methylation

Histone methylation occurs on different lysine and arginine residues that can be mono-, di-, and tri-methylated [29]. Histone methylation alters the chromatin status and thus influences the accessibility of protein factors to target DNA, which subsequently regulates gene expression. Histone methylation is established by histone methyltransferases (HMTs) and is removed by histone demethylases (HDM). HMTs mainly contain the SDG (SET-domain group) protein domain, which function to deposit methyl on specific histone lysine residues [30], while the two main classes of HDM, the JMJ (jumonji-C domain-containing protein) family and LSD1 (lysine-specific demethylases 1), act as erasers [31].

In plants, histone methylation mainly takes place in lysine (K) and arginine (R) residues, such as K4, K9, K27, K36, and K79, on the N-terminal tails of histones H3 and H4 [7]. Generally, H3K4me3 and H3K36me3 are reported to exert a function in transcriptional activation associated with open chromatin regions, while H3K27me3, H3K9me2, and H4R3 have been identified as repressive markers and hence, negative regulators of gene expression [7].

### 2.2. Histone Acetylation

Attachment of the acetyl group to histone tails is termed as acetylation. Histone acetylation leads to neutralization of the positive charge on the histone tail, weakening interactions between histone and DNA and thus resulting in the increased accessibility of transcription factors to DNA [21]. Histone acetylation is controlled by HATs (histone acetyltransferases) and HDACs (histone deacetylases) through a reversible process, where each separately acts as writers and erasers [32]. Four groups of HATs have been identified in Arabidopsis, including GNAT (GCN5-related N-acetyltransferase), MYST (MOZ, Ybf2/Sas2 and Tip60), CREB-binding protein (CBP)/p300, and TAF1 (TATA-binding protein-associated factor 1) [31]. HDACs in plants were classified into three families: RPD3/HDA1 (Reduced Potassium Dependence3/Histone DeAcetylase1) superfamily, SIR2 (Silent Information Regulator 2), and HD2 (Histone Deacetylase2) family [33]. Fourteen members were reported as HDACs in tomato, including SlHDA1-SlHDA9, SlHDT1-SlHDT3 (Sl HISTONE DEACETYLASE 1-3), and SlSRT1-SlSRT2 (Sl SIRTUIN 1-2) [34]. Overall, the histone acetylation marker is usually associated with gene activation, while deacetylation leads to a compact chromatin structure and the repression of gene expression [35].

On the other hand, histone phosphorylation strengthens the interaction with other types of histone modifications and is involved in DNA damage repair and chromatin conformation [36]. Histone ubiquitination is usually associated with transcriptional activation or repression, depending on the specific position in which it occurs. It also regulates DNA damage responses. Monoubiquitinated H2A (H2Aub) and H2B (H2Bub) are two of the most abundant ubiquitinated histones [37] in plants.

In this work, we focus primarily on the recent findings regarding histone methylation and the acetylation regulatory mechanisms related to the growth and development of rice and tomato, respectively (see Table 1). For other types of histone modifications, we refer the reader to [38,39].

## 3. Role of Histone Modifications in Improving Yield in Rice (*Oryza sativa* L.)

Rice is one of the major staple crops worldwide; thus, increasing rice yields is considered a great concern for food production. Recently, it has been shown that histone modifications make great contributions to rice growth, particularly by improving flowering efficiency, grain quality, and tolerance to adverse environments. Here we review the major histone modification mechanisms regulating flowering time, shoot and grain formation, and finally, response to abiotic stress (Figure 1).

### 3.1. Flowering

Flowering is the transition from vegetative growth to reproductive development, which directly influences reproduction efficiency and yield.

*Hd3a* [67] and *RFT1* [68] are two florigen genes dominating the flowering process and are under the control of the photoperiod. Additionally, a photoperiod-independent flowering pathway is mediated in rice by the B-type response regulator *Ehd1*. Flowering time, also termed heading date, is critical in determining yield and is modulated by histone modifications.

Recent work has revealed how H3K4me3 levels are regulated by OsTRX1 (TrithoRaX-like protein 1), hence subsequently affecting flowering time. OsTRX1 was found to interact with the transcription factor SIP1 (SDG723/OsTRX1/OsSET33 Interaction Protein 1) and was then recruited to its target *Ehd1*. Induction of *Ehd1* levels promotes flowering by upregulating the expression of florigen genes [69]. Loss of function mutants in *OsTRX1* and *SIP1* both led to reduced H3K4me3 levels at the *Ehd1* locus, thereby inhibiting *Ehd1* expression and resulting in a late heading date [40].

In addition, a positive correlation between H3K4me3 levels and flowering was also proved by another study. In fact, COMPASS (COMplex of Proteins Associated with Set1) complex, which is formed mainly by WD40 protein OsWDR5a and SDG 723 (SET Domain Group protein 723) was shown to positively regulate flowering and panicle branches by promoting H3K4me3 levels. OsWDR5a can promote *Ehd1* H3K4me3 deposition by binding to its promoter. As a result, plants with a lower expression of *OsWDR5a* showed reduced H3K4me3 levels at *Ehd1* and displayed unfavorable agronomic traits that reduce yield, such as delayed heading date, fewer secondary branches, and limited grain number, further confirming the positive correlation between H3K4me3 and flowering [41].

The transcription factor OsSUF4 was identified to interact with H3K36 methyltransferase SDG725. This interaction was required to recruit SDG725 to the promoter regions of the florigen genes *Hd3a* and *RFT1* for H3K36me3 deposition, thereby promoting gene induction and ultimately, flowering. Interestingly, loss of function mutant *suf4Ri-1* phenocopies *SDG725*-knockdown mutant *725Ri-1*, exhibiting a late flowering phenotype, regardless of the photoperiods. Furthermore, reduced H3K36me3 levels were detected in *suf4Ri-1* [42]. Altogether, this study preludes to a major role for H3K36me3 histone markers in activating florigen genes in rice.

In addition to histone methylation, histone acetylation also participates in controlling rice flowering. Histone deacetylase HDA703 regulates rice heading date through facilitating histone H4 deacetylation of *Ghd7*, a repressor of rice heading acting upstream of *Ehd1* [70]. As a result, overexpression of *HDA703* represses *Ghd7* expression, resembling the phenotype of *Ghd7*-silencing mutants, displaying accelerated growth rate throughout the whole period, including early heading [43].

Furthermore, histone variants also appear to be involved. Indeed, it was reported that OsINO80, an established chromatin-remodeling factor [71], interacts with histone variant H2A.Z to regulate flowering, seed germination, and reproductive efficiency through the gibberellin biosynthesis pathway. *OsINO80*-knockdown mutants showed decreased H2A.Z enrichments in GA biosynthesis genes such as *CPS1* (*ent-CoPalyl diphosphate Synthetase 1*) and *GA3ox2* (*Gibberellin 3-oxidase 2*), which led to impaired GA signaling and displayed a late flowering phenotype and reproductive development deficiency [44].

### 3.2. Shoot Development and Grain Formation

Apart from flowering efficiency, shoot and grain development are two additional traits in determining rice yield, which are also regulated by chromatin dynamics.

Enrichment of H3K27me3 at the promoter of genes associated with grain formation is correlated with distinct phenotypic responses. For instance, OsJMJ705, a major H3K27me3 demethylase in rice, can regulate shoot development through interaction with *WUSCHEL-related homebox (WOX)* genes, which are essential in controlling meristem development. Indeed, OsJMJ705 was found to mediate H3K27me3 removal from *WOX11* targets to activate their expression and promote shoot growth. As a result, genes involved in meristem identity that belong to the *Oryza sativa homeobox* (*OSH*) family, as well as transcripts encoding for chloroplast biogenesis and energy metabolism, were downregulated in *wox11* and *jmj705* mutants. Their expression profile was linked to various shoot growth defects, including decreased panicle length and reduced number of spikelets per panicle [45]. In accordance with the H3K27me3 role in gene repression, this study revealed a negative regulation through H3K27me3 on shoot growth.

Consistently, it was also found that the application of nitrogen fertilizer alters the genome-wide H3K27me3 pattern via NGR5 (Nitrogen-mediated tiller Growth Response 5)-dependent recruitment of PRC2 (Polycomb repressive complex 2) [46]. Recruitment of PRC2 led to altered H3K27me3 levels on loci encoding for branching-inhibitory genes and resulted in increased tiller number. This research also demonstrated that rice tillering and yield can be enhanced by improving NGR5 levels, without the excessive application of nitrogen fertilizer [46] 

Similarly, OsVIL2 (*Oryza sativa* VIN3-Like 2) can positively regulate grain size through chromatin remodeling of *OsCKX2 (CYTOKININ OXIDASEI DEHYDROGENASE2)* locus, which promotes cytokinin degradation and subsequently modulates panicle branches and grains number. In fact, H3K27me3 levels on the *OsCKX2* promoter region were increased in *OsVIL2* overexpressing lines, which indicates the repressed expression of *OsCKX2* and in turn, elevated levels of cytokinin, increased plant biomass, including increased tiller and spikelet number, and ultimately, higher yield [47]. This study demonstrated a positive role of H3K27me3 in regulating grain number by repressing negative regulators of the process.

Opposite to the general effect of H3K27me3, enhanced H3K4me3 levels in the promoter of *OsMADS (Os Mcm1, Agamous, Deficiens, Srf)*, which modulates grain size in rice, led to wider grains and increased yield [48].

Together with histone methylation, histone acetylation also functions to regulate grain size. Ubiquitin receptor HDR3 (Homolog of Da1 on Rice chromosome 3) facilitated H3 and H4 acetylation by stabilizing HAT GW6a (Grain Weight 6a), resulting in larger grain size and enhanced yield of rice [49].

### 3.3. Abiotic Stress

In rice, chromatin dynamics also appear to be critical for resistance to abiotic stress [50,53]. In one study, six histone markers (H3K4me3, H3K27me3, H4K12ac, H3K9ac, H3K27ac, and H3K36me3) were investigated in response to salt stress, demonstrating extensive yet precise gene regulatory effects through histone modifications [72].

Interestingly, AGO2 (ARGONAUTE2), despite its function in the small-RNA directed gene silencing, was found to optimize cytokinin distribution in shoots and roots, thereby increasing salt tolerance in rice by altering *BG3* (*BIG GRAIN3*) histone methylation levels and inducing its expression [73]. Histone deacetylase OsHDA710, which belongs to the HDAC RPD3/HDA1 family, also negatively controls salt tolerance by reducing the expression of stress responsive genes, including *OsLEA3* (*Os Late Embryogenesis Abundant protein3*), *OsABI5 (Os ABA Insensitive 5)*, and *OsbZIP72* (*Os Basic leucine ZIPPER 72*) [50].

IDS1 (INDETERMINATE SPIKELET1) was identified as a HDAC recruiter in the salinity stress response. IDS1 encodes for the apetala2/ethylene response factor, and negatively regulates rice salt tolerance by interacting with and recruiting histone deacetylase HDA1, resulting in the repression of key salt stress-responsive genes, including *LEA1* and *SOS1* (*Salt Overly Sensitive1*). Consistently, mutant lines *ids1-1* showed stronger tolerance and higher survival rate under NaCl treatment [51]. Plants suffering from drought stress can also shape their adaptation through histone modification mediated mechanisms. SDG708 (SET domain group protein 708), a H3K36 methyltransferase, was reported to improve the drought resistance of rice by preventing water loss and promoting stomatal closure [52]. OsJMJ703 catalyzes histone demethylation in the context of drought stress. In detail, the knockdown of *OsJMJ703* increases H3K4me3 levels, leading to accelerated flowering and enhanced tolerance in drought treatment. By contrast, the overexpression of *OsJMJ703* causes hypersensitivity to drought stress [53].

Altogether, these studies demonstrated that various types of epigenetic marks determine the very fine and robust regulation of gene expression primarily during plant adaptation. This suggests the use of epigenetic approaches in generating climate-smart rice plants and ensuring crop safety.

## 4. Role of Histone Modifications in Improving Fruit Quality in Tomato (*Solanum lycopersicum*)

Tomato is one of the world’s most economically important plants and favorable fruits. Up until now, epigenetic studies on tomato have primarily focused on increasing fruit ripening and quality, as well as improving its resistance to abiotic stress. Here, we focus on the major histone modifications and chromatin modifiers that affect tomato development, as well as its tolerance to environmental changes (Figure 2).

### 4.1. Fruit Ripening

In the context of fruit ripening, tomato has served as a model plant due to its features such as a short life cycle and its well-annotated genome [74]. Fruit ripening marks the terminal stage of its development and directly affects quality and flavor [75]. By analyzing the fruit ENCODE data, a sequenced reference database of seven climacteric fruit species [76], it was found that in immature tomato fruit, H3K27me3 was enriched primarily at the ethylene biosynthesis genes, while the same mark was absent from these regions upon fruit ripening. This suggests that H3K27me3 acts as a negative regulator of the ripening process (Figure 2) [77].

In accordance, histone demethylase SlJMJ6 was demonstrated to promote tomato fruit ripening by facilitating H3K27me3 demethylation, as overexpression of *SlJMJ6* led to activated expression of ripening-related genes and an acceleration of the tomato fruit ripening process [54].

LHP1,a component of PRC1 (Polycomb Repressive Complex 1), presents two variants, SlLHP1a and SlLHP1b, in tomato. SlLHP1b delays fruit softening and ethylene accumulation through colocalization with H3K27me3, and the overexpression of *SlLHP1b* downregulated ripening-related genes. This finding not only revealed the epigenetic mechanisms of how PcG (Polycomb Group) proteins regulate fruit ripening, but also strengthened the negative role of H3K27me3 [66].

In addition to that, transcriptional regulators such as NF-Y (Nuclear factor Y) are essential in flavonoid biosynthesis for forming NF-Y complexes and binding the CCAAT box in the promoter of the *CHS1* (*CHALCONE SYNTHASE1*) gene [78]. The low expression of *NF-YB* genes led to increased levels of H3K27me3 at the *CHS1* locus and subsequently, to reduced expression of *CHS1*, inhibiting the accumulation of flavonoid, and thus resulting in fruits with pink color and colorless peels [56]. Conversely, SlJMJ7 negatively regulates tomato fruit ripening by removing H3K4me3 methylation, which subsequently leads to the downregulation of ripening-related genes involved in ethylene biosynthesis [57]. These examples indicate that in tomato, the action of JMJs can be related to the activation or repression of gene expression, based on the type of histone methylation being removed, prompting future deeper investigations regarding the mechanisms of recognition of the histone marks.

Tomato fruit ripening is also modulated by histone acetylation (Figure 2). RNAi lines for *SlHDA3* showed early ripening, increased accumulation of carotenoid, and improved ethylene production, correlated with the activated expression of ethylene biosynthesis genes (*ACSs*, *ACOs*), as well as ripening-related genes, including *E4*, *E8*, *PG*, *Pti4*, and *LOXB* [63]. Similarly, the histone deacetylase SlHDT1 negatively regulates ripening by repressing carotenoid accumulation and ethylene biosynthesis. *SlHDT1-*RNAi lines showed early ripening, providing evidence of increased histone H3 acetylation levels correlating with fruit development [59]. SlERF.F12, a member of the ERF.F (Ethylene response factor. F) subfamily, is involved in delaying ripening by recruiting co-repressor TPL2 (TOPLESS 2) and HDA1/HDA3 (histone deacetylases 1/3) to their associated genes. As a result, H3K9Ac and H3K27Ac levels at promoter regions are decreased, leading to repressed transcription [60].

The transition from flower to fruit, also termed fruit set, is a critical shift in determining crop yield [79]. Differentially expressed genes (DEGs) were associated with the presence of H3K9ac or H3K4me3 markers during the flower-to-fruit transition. Enrichment in H3K9ac and/or H3K4me3, but low association with H3K27me3, was spotted in the same gene sets [60]. Besides the fruit set process, histone methylation also involves tomato flavor ester biosynthesis. *NOR* (*NON RIPENING*) positively regulates tomato fruit ripening, and *SlAAT1* (*Sl Alcohol AcylTransferases1*) dominates ester synthesis. A recent study showed that H3K4me3 in the *NOR* and *SlAAT1* loci increased, while H3K27me3 marks in these loci were removed as the fruits ripened. NOR activates *SlAAT1* transcription and contributes to volatile ester production in tomato. The binding site of NOR on the *SlAAT1* promoter is more accessible during fruit ripening [61]

### 4.2. Abiotic Stress

Tomato histone H3 lysine methyltransferases SDG (Set Domain Group) 33 and SDG34 were also studied to decipher the involvement of histone methylation in relation to drought response. Single mutants of *sdg33* and *sdg34* showed enhanced tolerance to drought, while double mutant *sdg33sdg34* showed even higher resistance, indicating an additive effect. Phenotypic responses were correlated with changes of H3K4me3 and H3K36me3 deposition at the target genes, with *sdg33sdg34* mutants exhibiting reduced methylation levels compared to the wild type [62]. Apart from its role in fruit ripening, NF-Y transcriptional regulators were also identified to respond to osmotic stress [80], primarily by mediating H3K27me3 levels at the target genes [75].

In a recent work, *SlHDA3* was reported to be involved in stress responses. *SlHDA3* RNAi lines showed shorter hypocotyl and root length, earlier yellowing and rolling leaves, and faster degradation of chlorophyll compared to wild type plants when exposed to drought, indicating that SlHDA3 might work as a positive regulator of abiotic stress tolerance [63]. Based on similar mechanisms, *SlHDA5*-RNAi lines exhibited less tolerance to salt and drought stresses, with chlorophyll in leaves degrading earlier and wilting faster in stressed conditions [64]. Likewise, *SlHDA1*-RNAi lines also showed decreased expression of genes that encode defensive proteins and demonstrated reduced capacity to withstand drought and salinity pressures [61]. These findings emphasize the critical role of HDACs in safeguarding tomato plants against adverse effects due to water scarcity and high salinity.

In addition to salt and drought stress, histone methylation also modulates tomato resistance to heavy metal, such as Cd (cadmium). Overexpression of the histone demethylase *SlJMJ524* exhibited enhanced tolerance to cadmium at the adult stage, showing lower Cd uptake and accumulation compared to WT plants [81]. Conversely, the overexpression of *SlJMJ4* promotes ABA-induced leaf senescence by decreasing H3K27me3 levels of ABA synthesis-related genes *SlNAP2 (Sl NAC domain-containing Protein 2)*, *SlORE1 (Sl ORESARA1)*, and *SlNCED3 (Sl Nine-Cis Epoxycarotenoid Dioxygenase 3)*, as well as activating their expression. SlJMJ4 increased plants sensitivity to ABA by binding to key genes related to ABA synthesis and signaling [66]. These findings not only support the notion that H3K27me3 negatively regulates stress associated gene expression, but also reveal a variety of processes that are affected by histone modifications. Chromatin markers can also be manipulated to improve tomato tolerance to low temperatures (0–12℃). Indeed, plant regulator coronatine was applied to enhance tomato’s chilling resistance by promoting H3K4me3 modifications and upregulating the expression of chilling responsive genes [82].

Overall, more histone modification-related components should be the target of molecular designs for tomato breeding.

## 5. Epigenetic Engineering in Crop Improvement

Epigenetic engineering is the directed editing of epigenetic marks at specific loci. ZFN (zinc finger nucleases) and TALENs (transcription activator-like effector nucleases) were first applied as epi-genome engineering tools, followed by the identification of CRISPR (Clustered Regularly Interspaced Short Palindromic Repeats) and its associated Cas (endonuclease) as one more advanced and powerful tool [83]. The utilization of deactivated Cas9 (dCas9) is prevalent due to its capacity to bind to the targeted area without including cleavage. This is achieved by using an enzymatically inactive version of the Cas9 protein. dCas9 can be fused with epigenetic modifiers and therefore initiate epigenetic regulation that causes transgenerational inherited phenotypic changes.

### Epi-Breeding and CRISPR-Cas9 Applications

The term epi-breeding refers to the application of epigenetic variation in crop amelioration [84,85]. Heritable variants of epigenetic markers are termed “epialleles”, and they can be adopted in epi-breeding and in making contributions to crop development as they influence agronomic traits [86,87,88]. The identification and application of epialleles with phenotypic variation has the potential to further benefit crop production, considering that genetic diversity has been overexploited by intensive breeding in long-term agricultural practices. DNA methylation, in particular, is partially heritable, indicating that agronomically important traits, such as seed dormancy, flowering time, and yield, can be impacted by modifications in DNA methylation status [89,90,91]. A more detailed description of the approaches and the potential of epigenetic variation for breeding can be referenced in [92,93]. Epialleles appear both naturally and artificially. In rice, several epialleles were identified which displayed altered agronomic traits. For example, *epi-d1* is an epiallele caused by the epigenetic silencing of *DWARF1* [94], showing a metastable dwarf phenotype. Upregulation of *OsSPL14*, *(Squamosa Promoter binding protein-Like 14)*, which is usually affected by microRNA excision, led to larger panicle branching and a higher yield in rice [95].

In tomato, it was demonstrated that naturally produced epialleles contained an altered accumulation of vitamin E in the fruits [96]. Specifically, the *cnr (colorless non-ripening*) mutation caused hypermethylation in the promoter of the SBP-box transcription factor and led to a delay in fruit ripening [97]. With the continuous discovery of naturally occurred epialleles and the generation of artificial ones, it will become more applicable to adopt epi-breeding in agriculture practices in the near future.

In this context, CRISPR/Cas9 has revolutionized genome editing and made epigenetic modification faster, easier, and cheaper [98]. CRISPR-Cas9 offers a site-specific modification of epigenetic markers, which may result in a significant improvement in plant fruit quality and resistance to various stresses, further confirming the potential within epi-breeding approaches [88,99]. For example, histone-modifying enzymes regulating stress tolerance were engineered to improve salt and/or drought resistance in transgenic plants. CRISPR/Cas9 was conducted on rice to generate the *HDA710* knockout mutant, which displayed stronger resistance to salt and ABA [50]. The same approach was also used to investigate the function of *OsHDA710* in callus formation of the mature rice embryo, which revealed that *Oshda710* showed impaired callus formation. By assessing CRISPR/Cas9, an epiallele of *JMJ705* was generated to confirm its role in regulating starvation stress. Similarly, *Ossnrk1a1* mutants were also generated to study their association with *JMJ705* [100], confirming CRISPR/Cas9 as highly effective in generating mutants for functional genetics. CRISPR/Cas9 approaches were also applied in studies related to agronomic traits, such as plant height. To investigate the role of SE1 (Silencing element of EUI1) interacting complex in processes involving histone deacetylation and H3K27me3 methylation of *EUI1* (*ELONGATED UPPERMOST INTERNODE1*) chromatin, a mutant for *OsVAL2*, one of the components of the SE1-associated complex, was generated using CRISPR/Cas9. All the CRISPR *Osval2* mutants showed increased expression of *EUI1* and displayed reduced plant height [101], proving that SE1-interacting complex is required for its repression, leading to elongated panicles, which benefit seed production.

CRISPR/Cas9 has also been applied to tomato fruit ripening studies. A tomato double mutant of the histone variant *Slh2a.z* was generated by gene-editing. *Slh2a.z* fruits exhibited reduced fresh weight and increased carotenoid content. Correlated with the loss of the H2A.Z variant, the expression of genes involved in carotenoid biosynthesis was significantly upregulated. This study provided evidence that epigenetic regulation via histone variants can also affect tomato ripening [102]. In addition, the *NONRIPENING* (*NOR*) deficient mutant was generated by CRISPR/Cas9, which showed reduced expression of *SlAAT1* and lower volatile ester production in ripening tomato fruits though its governing of *SlAAT1* histone methylation [73].

In summary, CRISPR/Cas9 represents an emerging gene-editing tool that also allows the targeting of the specific loci of epigenetic marks. Currently, the CRISPR-Cas9 approach shows equivalent efficiency compared to other genome editing methods, but it is an easier and less expensive tool, and it can target several regions (multiplex) within the same organism. CRISPR-Cas9 has already been extensively used in genome editing and is quickly becoming the most common technique to modify histone modifications in plants [103]. Several crop genomes, including maize, rice, cotton, potato, tomato, soybean, and sweet orange, have already been efficiently edited using CRISPR-Cas9 [99,103,104,105]. Moreover, epigenome editing mediated by CRISPR-Cas9 results in a gradual and proportional effect on the binding of epigenetic marks, and it has less dramatic off-target activity in comparison with genome editing methods. All these strategies work towards the same and the final goal of obtaining crops with higher yields or improved fruit quality in order to meet human demands.

## 6. Conclusions and Future Perspectives

In this review, we looked at the epigenetic mechanisms, with a focus on histone modifications contributing to rice and tomato growth and development, highlighting new approaches to improve the crop yield and fruit quality of the most dependable staple crops and vegetables in the world.

Here, we discussed how histone methylation and acetylation were involved in increasing plant flowering efficiency, fruit quality, and resistance to stress. Other types of histone modifications, such as phosphorylation and ubiquitination, which could potentially offer more solutions towards crop improvement, are still being investigated. So far, the generation of epialleles through epigenome editing technology is primarily intended for investigating the function of individual genes in specific regulatory mechanisms, rather than in developing crops for breeding purposes.

The hesitance towards the use of epigenetically modified crops may be attributed to concerns regarding their safety, as well as lack of awareness. However, it should be noted that breeding materials generated through this approach have the potential to significantly enhance food security, provided that the underlying mechanisms are thoroughly understood, and appropriate safety measures are implemented.

Epi-breeding can provide alternative and valuable methods for crop improvement, especially under the context that genetic variations have been overexploited by long-standing domestication and excessive breeding.

One of the limitations of epi-breeding resides in the lack of transgenerational inheritance of edited or modified histone markers, posing a challenge to its practical application. However, such a disadvantage could provide an opportunity to seek solutions to counteract the spreading of so called “genetic” contamination by ensuring that seeds from the same generation are employed.

With an increasing number of studies revealing the epigenetic mechanisms to improve crops and the function of the epialleles generated, epi-breeding may become a more practical way to contribute to food security under the trending climate change.

## Figures and Tables

**Figure 1 plants-12-01712-f001:**
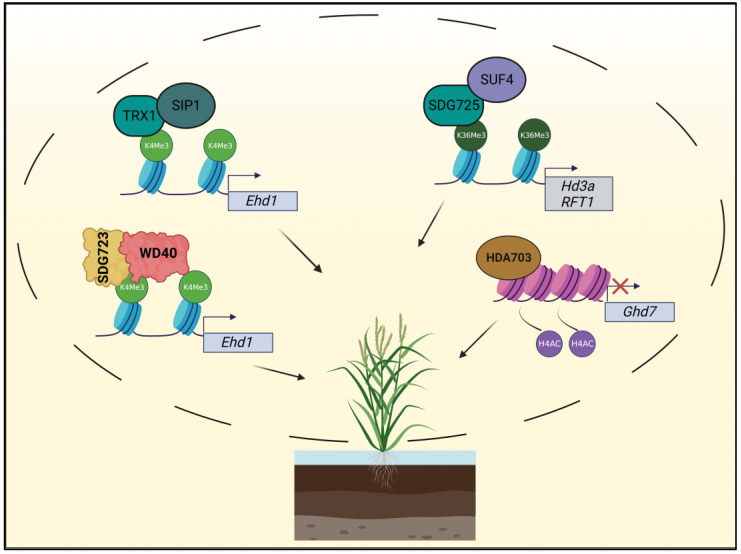
Overview of the histone modifications regulating flowering initiation in rice: TRX1 (TRITHORAX-like protein 1) interacts with SIP1 (SDG723/OsTRX1/OsSET33 Interaction Protein 1) to promote H3K4me3 deposition on the *Ehd1* (*Early heading date 1*) locus, thereby inducing flowering; the COMPASS complex that includes WD40 and SDG723 (Set Domain Group protein 723) positively induces flowering and panicle branches by increasing H3K4me3 levels on *Ehd1* promoter. SUF4 (SUppresor of Fri 4) mediates H3K36me3 on *Hd3a* (*Heading date 3a*) and *RFT1* (*Rice Flowering Time Locus T1*) promoters by interacting with histone methyltransferase SDG725. Conversely, histone deacetylase HDA703 deacetylates histone H4 (see histones in purple) on the *Ghd7* (*Grain number, plant height, and heading date7*) locus, thereby promoting the transcriptional activation of *Ehd1*.

**Figure 2 plants-12-01712-f002:**
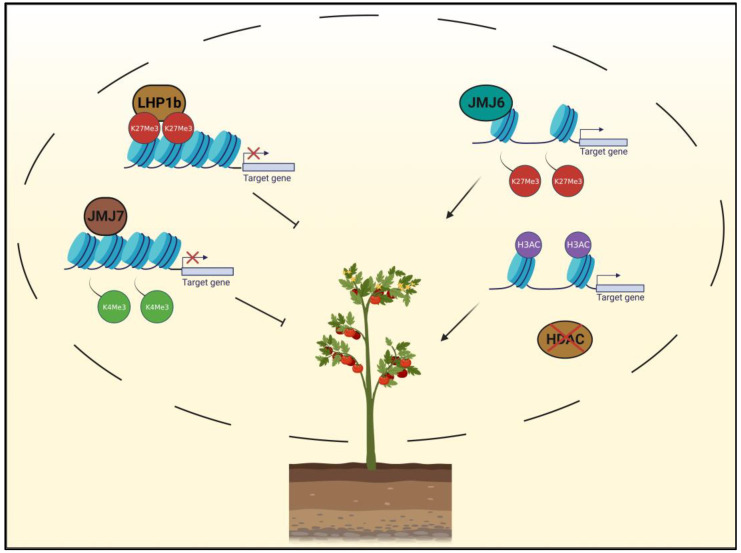
Tomato fruit ripening depends on chromatin positive and negative regulators of growth: LHP1b (Like Heterochromatin Protein 1b) delays fruit softening and ripening by accumulating H3K27me3 levels on target genes; histone demethylase JMJ (JUMONJI) 7 removes H3K4me3 markers on promoter regions, thereby reducing the expression of genes involved in ethylene biosynthesis. Conversely, JMJ6 reduces H3K27me3 levels on the same genes and induces their expression. Similarly, removal of histone deacetylases HDT1 promotes histone acetylation of genes involved in carotenoid accumulation and ethylene biosynthesis.

**Table 1 plants-12-01712-t001:** List of histone modification markers and the associated physiological processes in rice and tomato.

Species	Physiological Process	Epigenetic Mark	Associated Genes	Effect of Regulation	Reference
Rice (*Oryza sativa*)	Flowering	H3K4me3	*OsTRX1*, *SIP1*, *Ehd1*	Promotes flowering	[40]
			*OsWDR5a*, *Ehd1*	Promotes heading date and secondary branch growth	[41]
		H3K36me3	*OsSDG725*, *OsSUF4*, *Ehd1*	Promotes flowering	[42]
		De-Acetylation (H4K8, H4K12)	*OsHDA703*, *Ghd7*, *Ehd1*	Promotes flowering	[43]
		H2A.Z	*OsINO80*, *OsCPS1*, *OsGA3OX2*	Promotes flowering and reproductive efficiency	[44]
	Grain size and quality	H3K27me3	*OsJMJ705*, *OsWOX11*	Represses shoot growth	[45]
			*OsNGR5*, *OsPRC2*	Promotes tiller number	[46]
			*OsVIL2*, *OsCKX2*	Promotes cytokinin synthesis and plant biomass	[47]
		H3K4me3	*OsMADs*	Promotes grain size and yield	[48]
		Acetylation (H3 and H4)	*OsHDR3*, *GW6a*	Promotes grain size and yield	[49]
	Abiotic stress	Acetylation (H4K5, H4K16, H3K9)	*OsHDA710*, *OsLEA3*, *OsABI5*	Increases salt tolerance	[50]
		Acetylation (H3)	*OsIDS1*, *OsLEA1*, *OsSOS1*	Increases salt tolerance	[51]
		H3K36me3	*OsSDG708*	Increases drought resistance	[52]
		H3K4me3	*OsJMJ703*	Increases drought tolerance	[53]
Tomato (*Solanum lycopersicum*)	Fruit ripening	H3K27me3	*SlJMJ6*	Represses fruit ripening	[54]
			*SlLHP1b*	Represses fruit softening and ethylene accumulation	[55]
			*SlNF-YB*, *SlCHS1*	Represses flavonoid accumulation	[56]
		H3K4me3	*SlJMJ7*	Promotes ethylene biosynthesis	[57]
		Acetylation	*SlHDA3*, *SlACSs*, *SlE4*, *SlLOX8*	Promotes ripening and carotenoid accumulation	[58]
		Acetylation (H3)	*SlHDT1*	Promotes carotenoid accumulation and ethylene biosynthesis	[59]
		Acetylation (H3K9, H3K27)	*SlERF.F12*, *SlTPL2*, *SlHDA1/HDA3*	Promotes fruit ripening	[60]
		H3K4me3	*SlNOR*, *SlAAT1*	Promotes fruit ripening and ester synthesis	[61]
		H3K27me3	*SlNOR*, *SlAAT1*	Represses fruit ripening and ester synthesis	[61]
	Abiotic stress	H3K4me3 H3K36me3	*SlSDG33/34*	Promotes drought tolerance	[62]
		De-Acetylation	*SlHDA3*	Promotes drought resistance	[63]
			*SlHDA5*	Promotes salt and drought resistance	[64]
			*SlHDA1*	Promotes salt and drought resistance	[65]
		H3K27me3	*SlJMJ4*	Represses plant sensitivity to ABA	[66]

## Data Availability

Not applicable.

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
