# Peer review of "Unlocking the Secret to Higher Crop Yield: The Potential for Histone Modifications"

_plants, 2023, doi:10.3390/plants12081712_

Round 1

Reviewer 1 Report

The modification of histones through methylation and acetylation for controlling gene expression heritable but without changes in DNA sequences for important flowering, yield components and tolerance to abiotic stresses in rice and tomato has been extensively reviewed. The creation of new epialleles through mutation and genome editing has also been reviewed.  It is suggested to also review the methodology for the characterization of the epigenetic changes through histone modifications and DNA binding for some loci. The review article is well-written in scientific English without any mistakes.  

Author Response

The modification of histones through methylation and acetylation for controlling gene expression heritable but without changes in DNA sequences for important flowering, yield components and tolerance to abiotic stresses in rice and tomato has been extensively reviewed. The creation of new epialleles through mutation and genome editing has also been reviewed.  It is suggested to also review the methodology for the characterization of the epigenetic changes through histone modifications and DNA binding for some loci. The review article is well-written in scientific English without any mistakes.  

We thank the reviewer for the comment: we have extended the introduction on histone modifications and added a new paragraph where we briefly introduced the most used methodologies like ChIP and the latest technologies to study chromatin organization and topology (see revised  2. "Histone modifications").  

Reviewer 2 Report

This review article entitled “Unlocking the secret to higher crop yield: the potential for Histone Modifications”. In this review article, the authors have summarized the recent advances of epigenetic regulatory mechanisms underlying crop flowering efficiency, fruit quality, and adaption to environmental stimuli, especially to abiotic stress, to ensure crop yield improvement. Moreover, they summarized and discussed the applications of epigenetic approaches in crop improvement and breeding.

The topic is interesting and informative. The manuscript needed to be revised before it could be published. Before recommending this article for publication, some shortcomings should be resolved.

The authors should improve the overall English of the manuscript.

Many formatting mistakes have been found. I suggest authors review the whole manuscript carefully and correct all the mistakes.

The scientific names of the species and the names of the genes must be italicized in the manuscript.

The authors should fully explain the abbreviations during the first mention in the abstract and introduction.

Line 18: quality, and  

Line 18: stimuli, especially

In the abstract, please use “summarize” or “summarized”.

Line 26: seasonal change and circadian clock

Line 30: development, and

Line 34: RNAs, and

Line 41: referred to?

Line 43: glycosylation, and

Line 47: yield [12,13],

Line 50: heat and freezing

Line 82: residues, such as

Line 83: K36, and K79

Line 101-103: This paragraph is too short; please merge it with another paragraph.

Line 114: Please write the full scientific names, such as Oryza sativa L.

Line 118: Please give a space, Fig.1

What’s Ehd1?

How could authors start a sentence with the name of the gene?

Line 303: metal, such as

Lines 308-309: (SlNAP2, SlORE1, and SlNCED3)

Line 395: 6. 

Line 412: improvement, especially

Author Response

This review article entitled “Unlocking the secret to higher crop yield: the potential for Histone Modifications”. In this review article, the authors have summarized the recent advances of epigenetic regulatory mechanisms underlying crop flowering efficiency, fruit quality, and adaption to environmental stimuli, especially to abiotic stress, to ensure crop yield improvement. Moreover, they summarized and discussed the applications of epigenetic approaches in crop improvement and breeding.

The topic is interesting and informative. The manuscript needed to be revised before it could be published. Before recommending this article for publication, some shortcomings should be resolved.

The authors should improve the overall English of the manuscript.

We have extensively reviewed the manuscript and revised the overall English.

Many formatting mistakes have been found. I suggest authors review the whole manuscript carefully and correct all the mistakes.

All the formatting mistakes have been rectified, thank you.

The scientific names of the species and the names of the genes must be italicized in the manuscript.

All the scientific gene names are now in Italic. However, protein names are not, according to general nomenclature. Additionally, rice gene names follow a different nomenclature.

The authors should fully explain the abbreviations during the first mention in the abstract and introduction.

All the abbreviations have been fully explained when mentioned the first time in the text.

Line 18: quality, and  

Changed it, thank you.

Line 18: stimuli, especially

Changed it, thank you.

In the abstract, please use “summarize” or “summarized”.

In the abstract we now only use summarize.

Line 26: seasonal change and circadian clock

Changed it, thank you.

Line 30: development, and

Changed it, thank you.

Line 34: RNAs, and

Changed it, thank you.

Line 41: referred to?

It is referred to the references in brackets.

Line 43: glycosylation, and

Changed it, thank you.

Line 47: yield [12,13],

Changed it, thank you.

Line 50: heat and freezing

Changed it, thank you.

Line 82: residues, such as

Changed it, thank you.

Line 83: K36, and K79

Changed it, thank you.

Line 101-103: This paragraph is too short; please merge it with another paragraph.

Paragraphs merged.

Line 114: Please write the full scientific names, such as Oryza sativa L.

We added the full scientific name, thank you.

Line 118: Please give a space, Fig.1

Done it, thanks.

What’s Ehd1?

Many thanks for your question: Ehd1 is B-type response regulator that promotes flowering in rice through a photoperiod-independent pathway (see also paragraph 3.1).

How could authors start a sentence with the name of the gene?

We changed the start of the sentence with “Induction of Ehd1 levels..”

Line 303: metal, such as

Changed it, thank you.

Lines 308-309: (SlNAP2, SlORE1, and SlNCED3)

Changed it, thank you.

Line 395: 6.

Changed it. 

Line 412: improvement, especially

Changed it.

Reviewer 3 Report

It is an interesting manuscript to describe the potential for the applications of epigenetic approaches in crop improvement and breeding. The manuscript is mostly well written. Apart from that, I have the following suggestions for the author to consider in further improving the manuscript.

1.      Lowercase the first letter of Histone Modifications in the title.

2.      Line 82-86: Please provide more details about histone methylation on arginine residues.

3.      Line 114: Lowercase the first letter of Histone.

4.      Line 134-141: Please rephrase this part.

5.      Line 195-200: Please provide more details about callus formation in relation to crop yield.

6.      Table 1: Please provide the detail of epigenetic mark of acetylation and so on.

7.      Figure 1 and 2: Please provide the histone methylation takes place at lysine of histone H3.

8.      References: The journal names should be in a uniform format. 

Author Response

It is an interesting manuscript to describe the potential for the applications of epigenetic approaches in crop improvement and breeding. The manuscript is mostly well written. Apart from that, I have the following suggestions for the author to consider in further improving the manuscript.

We thank the reviewer for the positive comments. We have taken on board their suggestions.

  1. Lowercase the first letter of Histone Modifications in the title.

Changed it, thank you.

  1. Line 82-86: Please provide more details about histone methylation on arginine residues.

We have included the arginine residues involved in histone methylation (see revised paragraph 2.1).

  1. Line 195-200:Please provide more details about callus formation in relation to crop yield.

We thank the reviewer for the comment. Since the relevance to crop yield is minor, we have deleted that sentence.

  1. Table 1: Please provide the detail of epigenetic mark of acetylation and so on.

We have added the information according to the literature, thank you.

  1. Figure 1 and 2: Please provide the histone methylation takes place at lysine of histone H3.

Except for one example, all the modifications reported in the figures occurred on H3. We have now made it clear in the figure legend (see revised legend for figure 1) .

  1. References: The journal names should be in a uniform format.

We have used Mendeley as reference manager program that has generated the bibliography based on Plants Citation style.

Reviewer 4 Report

Evaluation of the review article entitled:

Unlocking the secret to higher crop yield: the potential for Histone Modifications

 Summary

The article reviews general aspects of epigenetics and more specifically histone modifications and the effects of histone methylation and acetylation on crop (rice and tomato) development, seed formation and fruit maturation.

Two additional chapters deal briefly with Epigenetic engineering in crop improvement and Epi-breeding and CRISPR-Cas9 applications, respectively.

General comment

The review article is dedicated to a subject of interest. While it is well written and ordered at the beginning (sections 1 and 2), reading sections 3, 4 and 6 is more difficult. This is unfortunate, because the subject in these sections is interesting and novel, and the confuse writing will reduce the number of readers and the interest of the article. Please re-write these sections in a more ordered and reasonable way.

Comments by sections

2.2. Histone acetylation:

Please verify or correct:

Histone acetylation is controlled by HATs (histone acetyltransferases) and HDACs (histone deacetylases) through a reversible process, where each separately acts as readers and erasers[22].

In this sentence it should be writers, instead of readers.

3. Role of Histone modifications in improving yield in rice (Oryza sativa)

This section should be re-written to offer a more ordered text. It should be divided in sub-sections with each case separated from the others, and all of them preceded by a general introductory sub-section. Thus: Subsection 1: Introductory; Subsection 2: Flowering; Subsection 3: Seed number and seed size.

Please specify at the start of each sub-section (2 and 3) how many cases will be treated in this section and their main components. At least start a new paragraph each time a new case is presented, for example: Line 142 (start of OsSUF4), Line 149 (start of histone acetylation), Line 175 (start of NGR5), Line 182 (OsVIL2)…

 Apparently a section on grain size and grain filling starts on line 164. But the first lines of this new section state:

 “Enrichment of H3K27me3 at the promoter of genes associated with grain size is correlated with distinct phenotypic responses. For instance, OsJMJ705, a major H3K27me3 demethylase in rice, can regulate shoot development through interaction with WUSCHEL-related homebox genes. Indeed, OsJMJ705 was found to mediate H3K27me3 removal from WOX11 targets, which is essential to activate their expression and promote shoot growth. As a result, genes involved in meristem identity, chloroplast biogenesis and energy metabolism were downregulated in wox11 and jmj705 mutants that exhibited shoot growth defects such as reduced panicle length and reduced number of spikelet per panicle [39]. In accordance with H3K27me3 role on gene repression, this study revealed a negative regulation through H3K27me3 on grain forming.”

The following questions arise:

What are those genes associated with grain size?

What is, if any, their relation with shoot development?

As it is explained there is little relation of this case with grain development.

Please correct or explain:

“Similarly, OsVIL2 (Oryza sativa VIN3-LIKE 2) can positively regulate grain size through the chromatin remodeling of OsCKX2 (CYTOKININ OXIDASEI 183 DEHYDROGENASE2) locus, which promotes cytokinin degradation and subsequently modulates panicle branches and grains.”

Panicle branches and grains?

Number of grains? Grain size? Please give as much details as possible of the effect of OsCKX2 on grains.

Again:

“This study demonstrated a positive role of H3K27me3 in regulating grains forming by repressing negative regulators of the process.”

Number of grains? Grain size?...

4. Role of Histone modifications in improving fruit quality in tomato (Solanum 231 lycopersicum)

Similar to Sec. 3, this section may benefit from the division in sub-sections with an indication of the respective contents for any of them.

5. Epigenetic engineering in crop improvement

Please check if all references due are included in this section.

Author Response

The review article is dedicated to a subject of interest. While it is well written and ordered at the beginning (sections 1 and 2), reading sections 3, 4 and 6 is more difficult. This is unfortunate, because the subject in these sections is interesting and novel, and the confuse writing will reduce the number of readers and the interest of the article. Please re-write these sections in a more ordered and reasonable way.

We thank the reviewer for the constructive comments. We agree that sections 3-4 are difficult to follow, therefore we have divided them into four and three subparagraphs, respectively. We have also substantially revised section 6 accordingly (see revised version of the manuscript).

 Comments by sections

2.2. Histone acetylation:

Please verify or correct:

Histone acetylation is controlled by HATs (histone acetyltransferases) and HDACs (histone deacetylases) through a reversible process, where each separately acts as readers and erasers[22].

In this sentence it should be writers, instead of readers.

Changed it, thank you. 

  1. Role of Histone modifications in improving yield in rice (Oryza sativa)

This section should be re-written to offer a more ordered text. It should be divided in sub-sections with each case separated from the others, and all of them preceded by a general introductory sub-section. Thus: Subsection 1: Introductory; Subsection 2: Flowering; Subsection 3: Seed number and seed size.

Paragraph 3 is now divided into 4 subsections: 3. Introduction; 3.1 Flowering; 3.2 Shoot Development and grain formation; 3.3 Abiotic stress. In addition we have included a small paragraph at the beginning to introduce the following subsections.

Please specify at the start of each sub-section (2 and 3) how many cases will be treated in this section and their main components. At least start a new paragraph each time a new case is presented, for example: Line 142 (start of OsSUF4), Line 149 (start of histone acetylation), Line 175 (start of NGR5), Line 182 (OsVIL2)…

Thanks for the comment: now every case is dedicated to a new paragraph. 

 Apparently a section on grain size and grain filling starts on line 164. But the first lines of this new section state:

 “Enrichment of H3K27me3 at the promoter of genes associated with grain size is correlated with distinct phenotypic responses. For instance, OsJMJ705, a major H3K27me3 demethylase in rice, can regulate shoot development through interaction with WUSCHEL-related homebox genes. Indeed, OsJMJ705 was found to mediate H3K27me3 removal from WOX11 targets, which is essential to activate their expression and promote shoot growth. As a result, genes involved in meristem identity, chloroplast biogenesis and energy metabolism were downregulated in wox11 and jmj705 mutants that exhibited shoot growth defects such as reduced panicle length and reduced number of spikelet per panicle [39]. In accordance with H3K27me3 role on gene repression, this study revealed a negative regulation through H3K27me3 on grain forming.”

The following questions arise:

What are those genes associated with grain size?

Thank you. For your comment. We have included a short introduction to the text to avoid confusion. Regarding this specific paragraph, we actually talk about shoot and panicle development, which are correlated to yield. We have revised the sentence and included the gene class associated with the phenotype.

What is, if any, their relation with shoot development?

As it is explained there is little relation of this case with grain development.

Thank you for the comment. Here we have revise the text : shoot and panicle development and crop yield. In addition in the text we clarify that knock out of these genes lead to defects to shoot growth.

Please correct or explain:

“Similarly, OsVIL2 (Oryza sativa VIN3-LIKE 2) can positively regulate grain size through the chromatin remodeling of OsCKX2 (CYTOKININ OXIDASEI 183 DEHYDROGENASE2) locus, which promotes cytokinin degradation and subsequently modulates panicle branches and grains.”

Panicle branches and grains?

Number of grains? Grain size? Please give as much details as possible of the effect of OsCKX2 on grains.

Thank you for pointing this out. It is actually panicle branches and grain number, we have now added it to the text. Repressed expression of OsCKX2 leads to increased tiller numbers and spikelets in one plant.

Again:

“This study demonstrated a positive role of H3K27me3 in regulating grains forming by repressing negative regulators of the process.”

Number of grains? Grain size?...

Thank you for your comment. We have clarified the text by replacing grains forming” to “grain number.

  1. Role of Histone modifications in improving fruit quality in tomato (Solanum 231 lycopersicum)

Similar to Sec. 3, this section may benefit from the division in sub-sections with an indication of the respective contents for any of them.

Thank you for your comment: We have divided the paragraph in three subsections: 4. Introduction; 4.1 Fruit ripening; 4.2 Abiotic stress. 

  1. Epigenetic engineering in crop improvement

Please check if all references due are included in this section.

We have included new references in the text.

Round 2

Reviewer 2 Report

Thanks to the authors for incorporating all the suggested changes.

Author Response

Many thanks to the reviewer for the useful suggestions.

Reviewer 4 Report

The article has improved in this version. Nevertheless one of the questions raised in the previous round of review remains unanswered. Please make it clear now:

4. Role of Histone modifications in improving fruit quality in tomato (Solanum lycopersicum)

It seems to be some contradiction between:

“In accordance, histone demethylase SlJMJ6 was demonstrated to promote fruit ripening of tomato by facilitating H3K27me3 demethylation, as overexpression of SlJMJ6 led to activated expression of ripening related genes and an acceleration of tomato fruit ripening process [53].”

 And:

“Conversely, SlJMJ7 negatively regulates tomato fruit ripening by removing H3K4me3 methylation which subsequently leads to the downregulation of ripening-related genes involved in ethylene biosynthesis [15].”

From these two paragraphs it seems that there may be both an activation or a repression of the promoters by methylation depending on the methylation type. Please confirm and explain well this aspect.

Author Response

The article has improved in this version. Nevertheless one of the questions raised in the previous round of review remains unanswered. Please make it clear now:

  1. Role of Histone modifications in improving fruit quality in tomato (Solanum lycopersicum)

It seems to be some contradiction between:

“In accordance, histone demethylase SlJMJ6 was demonstrated to promote fruit ripening of tomato by facilitating H3K27me3 demethylation, as overexpression of SlJMJ6 led to activated expression of ripening related genes and an acceleration of tomato fruit ripening process [53].”

 And:

“Conversely, SlJMJ7 negatively regulates tomato fruit ripening by removing H3K4me3 methylation which subsequently leads to the downregulation of ripening-related genes involved in ethylene biosynthesis [15].”

From these two paragraphs it seems that there may be both an activation or a repression of the promoters by methylation depending on the methylation type. Please confirm and explain well this aspect.

Many thanks for this comment. Yes, the effect of the Sl JMJ action can change depending on the type of methylation being removed. To clarify this aspect we added the following sentence to the text "These examples indicate that in tomato the action of JMJs can be related to the activation or repression of gene expression based on the type of histone methylation being removed, leading, in the future, towards a deeper investigation on the mechanisms of recognition of the histone marks".